# CAR Cell-Derived Exosomes in Cancer Therapy: Biogenesis, Engineering Strategies and Antitumor Mechanisms

**DOI:** 10.3390/ijms26167890

**Published:** 2025-08-15

**Authors:** Chaohua Si, Yuanyuan Li, Yunwen Wang, Jianen Gao, Xu Ma

**Affiliations:** National Research Institute for Family Planning, Chinese Academy of Medical Sciences & Peking Union Medical College, Beijing 100000, China; pumcsch@pumc.edu.cn (C.S.); liyuanyuan@biols.ac.cn (Y.L.); wangyunwen@nrifp.org.cn (Y.W.)

**Keywords:** CAR cells-derived exosomes, tumor therapy, clinical translation, biogenesis, mechanism

## Abstract

Chimeric antigen receptor (CAR) cell therapy, encompassing CAR T, CAR NK, and CAR macrophage cells, demonstrates high efficacy in tumor treatment, conferring durable and effective responses, notably in hematologic malignancies. However, challenges persist in the manufacture of CAR cells, and treatment is associated with serious adverse events, notably cytokine release syndrome (CRS), a potentially life-threatening complication. Owing to the inherent properties of exosomes, CAR cell-derived exosomes offer distinct advantages in cancer therapeutics. CAR cells-derived exosomes retain the inherent tumor-killing function of the parent cells while also exhibiting key practical advantages, including wide availability, safety, and ease of storage and transport. Furthermore, CAR cell-derived exosomes can be combined with other tumor therapies; this combinatorial approach significantly enhances efficacy while reducing side effects. To accelerate the clinical translation of CAR cell-derived exosomes in tumor therapy, this paper reviews their biogenesis, engineering strategies, antitumor mechanisms and clinical evidence, including case studies of combination therapies with other antitumor modalities.

## 1. Introduction

Globally, cancer presents a significant health challenge, as annual incidence and mortality rates continue to increase [1]. Although surgery, radiotherapy, and chemotherapy represent the primary modalities for cancer treatment, they remain limited in efficacy, particularly for metastatic or recurrent cases [2,3]. Advances in cancer immunotherapy research have spurred progress in targeted therapy, enabling precision medicine to emerge as a more refined cancer treatment strategy with improved safety profiles [4]. Targeted antitumor therapy works by modulating the immune system to enhance the recognition and destruction of tumor cells, thereby inhibiting cancer progression [5]. A prominent antitumor cellular immunotherapy utilizes immune cells engineered to express receptors specific for tumor surface antigens, leading to the destruction of the targeted tumor cells [6]. A key application of gene editing is the engineering of immune cells to express chimeric antigen receptors (CAR), which confer precise tumor-targeting capabilities. Current research thus centers on CAR-based immunotherapies, notably CAR T cells, CAR NK cells, and CAR M [7]. Clinically approved CAR-T cell therapies have already demonstrated significant efficacy in the treatment of specific malignancies, including B-cell leukemia [8,9]. However, CAR cell therapy efficacy is limited by insufficient enrichment of CAR cells at the tumor site, a consequence of the immunosuppressive microenvironment [10]. Additionally, CAR T-cell therapy is associated with notable adverse effects, notably cytokine release syndrome (CRS) and on-target/off-tumor toxicity [11]. To overcome these obstacles, researchers are pursuing novel therapeutic modalities.

Exosomes are extracellular vesicles, approximately 30–150 nm in diameter, that are released following the fusion of multivesicular bodies (MVBs) with the plasma membrane [12]. Exosomes contain diverse biomolecules, such as nucleic acids, proteins, and lipids [13]. Exosomes facilitate intercellular exchange of substances to maintain cellular homeostasis and mitigate external stress. Research indicates their involvement in tumor progression, including processes such as metastasis, chemoresistance, angiogenesis, and apoptosis [14]. As research on CAR-T cell therapy advances, researchers recognize that CAR-T cell-derived exosomes retain most of the cells’ cytotoxic capabilities while offering unique advantages that address key limitations of CAR-T therapy itself [15,16]. For instance, CAR cell-derived exosomes present a safer alternative to CAR-T cell therapy in tumor treatment, notably avoiding the risk of cytokine release syndrome. Furthermore, compared to CAR-T cells, these exosomes overcome tumor microenvironment (TME) barriers to enhance therapeutic efficacy while offering superior storage and transport stability [17,18]. These advantages confer strong clinical application potential to CAR cell-derived exosomes.

Furthermore, CAR cell-derived exosomes function not only as inherent antitumor agents but also as novel delivery platforms for therapeutic molecules, enhancing treatment efficacy by targeting tumor cells [19]. Current research indicates no universal approach to tumor therapy exists; effective inhibition of tumor progression necessitates synergistic strategies combining multiple therapeutic modalities [20]. CAR cell-derived exosomes represent a potential adjunctive therapy that can be combined with established tumor treatments such as radiotherapy and chemotherapy, often yielding greater efficacy alongside reduced side effects [21]. In conclusion, this article reviews the roles, mechanisms, and recent advances in CAR-T-derived exosomes for tumor treatment. It further systematically summarizes the clinical applications and value of CAR-T-derived exosomes, aiming to accelerate their clinical translation in tumor therapy.

## 2. CAR-Cell Therapy in Cancer Progress

CAR cell therapy genetically engineers immune cells to target cancer cells. The three primary types—CAR-T, CAR-NK, and CAR-M—leverage T cells, natural killer cells, and macrophages, respectively. Next, we will summarize the current research status and progress of them in tumor research (Clinical Phase 1, CRS > 50%; Figure 1; Table 1).

### 2.1. CAR-T Cell Therapy

In immunocompetent cancer patients, tumor cells often evade immune surveillance through cancer immunoediting, establishing a tumor-promoting microenvironment [22]. Overcoming tumor immune escape mechanisms, such as tumor-associated antigen (TAA) deficiency and T-cell dysfunction, is therefore essential for effective cancer therapy [23,24,25]. Current approaches involve ex vivo T cell expansion/activation or genetic engineering, primarily encompassing three modalities: chimeric antigen receptor (CAR) T cell therapy, tumor-infiltrating lymphocyte (TIL) therapy, and T cell receptor-engineered (TCR) T cell therapy [26]. CAR T-cell therapy involves genetically engineering T cells to express synthetic receptors (CARs). These receptors comprise an extracellular antigen-binding domain (typically a single-chain variable fragment), a hinge region, a transmembrane domain, and intracellular signaling domains that activate cytotoxicity and co-stimulatory pathways upon antigen recognition [27,28]. These engineered T cells are MHC-independent [29].

### 2.2. Car-NK Cell Therapy

Due to their intrinsic ability to target tumors, natural killer (NK) cells represent a promising cancer immunotherapy, driving significant interest in CAR-NK cells [30,31]. Current clinical evidence for CAR NK-cell therapy primarily derives from hematologic malignancies. Insights from CAR T-cell research and NK cell biology may inform its application and challenges in solid tumors [32,33]. Current research indicates that CAR NK cells face major challenges in treating solid tumors, including poor tumor homing, inadequate infiltration, and failure to persist. Owing to deleterious factors within the tumor microenvironment (TME), such as soluble suppressors and hypoxia, infiltrated NK cells often lose their functional phenotype, compromising therapeutic efficacy [34,35,36,37,38]. Current CAR NK research aims to overcome these challenges by focusing on four main areas: First, enhancing NK cell infiltration into solid tumors [39,40,41]. Second, CAR-NK cell persistence correlates with patient outcomes [42,43]. Third, overcoming the suppressive tumor immune microenvironment (TME) [44,45,46]. Fourth, optimisation of NK cell-based CAR constructs.

### 2.3. Car-Macrophage Cell Therapy

Macrophages effectively infiltrate tumors, are abundant within the TME, and serve as key immunomodulators. Therefore, developing macrophage-based CAR therapies may be a promising approach to overcome major obstacles associated with CAR T/NK therapy for solid tumors. CAR T-cell and CAR NK-cell therapies have advanced significantly in hematological malignancies but have proven less effective in solid tumors, largely due to distinct biological differences; for example, antigenic targets remain poorly defined. Given the limitations of CAR T and NK cell therapies, CAR macrophages have emerged as alternatives owing to their unique advantages, primarily tumor infiltration capability within the TME. Unlike T cells and NK cells, macrophages are abundant in many tumors and secrete numerous cytokines at infiltration sites. For example, they constitute up to 50% of infiltrating cells in melanoma, colorectal cancer, and renal cell carcinoma [47] (Figure 2A).
ijms-26-07890-t001_Table 1Table 1Part of published clinical trials of CAR cell therapies.ProductClinical Trials IDStudy PhaseEdited GenesCancerTarget AntigensEfficacyToxicitiesRefs.TT52CAR19NCT04557436ICD52B-ALLCD19A total of 6 evaluable patients, 4 of whom (66.7%) had CR, received allo-HSCT consolidation.CRS: 100%;ICANS: 33.3%;GVHD: 33.3%;Cytopenias: 100%.[48]CTA101NCT04227015ICD52B-ALLCD19 and CD22A total of 6 evaluable patients, 5 of whom (66.7%) had CR/Cri.CRS: 100%;ICANS: none;GVHD: none;Cytopenias: 50%.[49]UCART19NCT02640209ICD52B-ALLCD197 evaluable children, 25 evaluable adults. 6/7 children (85.7%) CR/CRi, 12/25 adults (48%) CR/Cri.CRS: 90.5%;ICANS: 38.1%;GVHD: 9.5%;Cytopenias: 31.6%.[50]BE-CAR7NCT05397184ICD52 and CD7T-ALLCD7A total of 3 evaluable patients, 3 of whom (100%) had CR/Cri.CRS: 100%;ICANS: 33%;GVHD: 33%;Cytopenias: 100%.[51]GC027NCT04264078ICD7T-ALLCD7A total of 12 evaluable patients, 11 of whom (91.7%) had CR/Cri.CRS: 83.3%;ICANS: none;GVHD: NR;Cytopenias: NR.[52]CTX130NCT04502446ICD70ccRCCCD70A total of 16 evaluable patients, 1 of whom (6.2%) had CR/Cri.CRS: 50%;ICANS: none;GVHD: none;Cytopenias: NR.[53]

## 3. Comparison of CAR Cell-Derived Exosomes and CAR Cells in Cancer Therapy

### 3.1. Limitations of CAR Cell Therapy

#### 3.1.1. Tumor Microenvironment

The tumor microenvironment (TME) is the complex system in which tumor cells reside, comprising primarily tumor cells, immune cells, stromal cells, extracellular matrix, and cytokines [54]. Progressive disease leads to T cell exhaustion within the TME due to persistent tumor antigen exposure. This impairs antitumor function and can promote tumor progression, fostering an immunosuppressive microenvironment. TME components inhibit CAR cell function, compromising tumor therapy outcomes. For instance, TME components like myeloid-derived suppressor cells (MDSCs), tumor-associated macrophages (TAMs), and regulatory T cells (Tregs) can modulate CAR cell function through cytokine secretion. Moreover, oxygen concentration and nutrient imbalance suppressed CAR cell activity (Figure 2B).

#### 3.1.2. Lack of Specific Targets

Currently, all FDA-approved CAR cell therapies target B-spectrum markers. The lack of specific targets is a key reason CAR cell therapy for solid tumors lags behind its success in hematologic malignancies. CAR T-cell therapy targeting CD19 induces B-cell depletion; however, intravenous immunoglobulin (IVIG) supplementation can compensate for most functional deficits. However, this non-functionally specific antigen is virtually absent in solid tumors. Clinical trials of CAR T-cell therapy targeting the antigens MART1 and gp100 in melanoma patients demonstrated serious, occasionally irreversible adverse effects [55] (Figure 2C).

#### 3.1.3. Insufficiency of CAR Cell Expansion and Persistence

CAR cell expansion and persistence in vivo are crucial determinants of tumor therapy efficacy, especially in malignancies necessitating prolonged treatment. The expansion and persistence of CAR cells in vivo critically influence tumor therapy efficacy, particularly for malignancies requiring long-term treatment [56]. Furthermore, CAR cell expansion capacity and persistence depend directly on their intrinsic properties [57]. Preclinical and clinical evidence indicate that the co-stimulatory domain critically influences CAR T cell persistence in vivo, with CD28 domains conferring persistence for ~30 days and 4-1BB domains enabling persistence up to 168 days [58,59,60,61]. Repeated antigen exposure can cause administered CAR T cells to exhaust and undergo ACID. Additionally, CAR-T cell depletion may occur if antigen levels are too low to sustain activity after target cell clearance [62] (Figure 2D).

#### 3.1.4. Inefficiency of CAR Cell Trafficking and Infiltration

Post-administration, CAR cell delivery and tumor infiltration efficiency critically affect therapeutic efficacy. Abnormal expression of adhesion molecules in tumor vasculature impedes CAR cell migration, adhesion, and infiltration into tumor sites. The tumor microenvironment (TME) stroma, including cancer-associated fibroblasts (CAFs), impedes CAR cell entry into tumor sites. To overcome this challenge, numerous strategies aim to enhance CAR cell infiltration [63,64]. For example, combining CAR T-cell therapy targeting CAFs with another CAR T-cell approach, or expressing cytokine receptors that bind tumor-upregulated cytokines (Figure 2E).

#### 3.1.5. Down-Regulation or Absence of Target Antigen

Loss of target antigens represents a common mechanism of tumor resistance to CAR-T cell therapy. Due to the heterogeneity of solid tumors in target antigens, antigen loss or escape may significantly impact treatment efficacy. Among patients relapsing after CD19-targeted CAR T-cell therapy, approximately 25% exhibited CD19 downregulation. To address this, current efforts focus on modifying CAR structures to enhance sensitivity to low antigen expression, target alternative antigens, and more effectively induce antitumor immune responses [65] (Figure 2F).

#### 3.1.6. Host-Mediated Graft Rejection

Most approved CAR-T products are autologous, primarily because allogeneic alternatives carry a significant risk of graft rejection [66]. Pre-infusion lymphocyte depletion mitigates T-cell-mediated allograft rejection. In clinical trials, increasing fludarabine or cyclophosphamide dosage enhances lymphatic clearance, whereas engineering purine nucleoside analogue resistance into CAR T cells prolongs its duration [51,67]. Editing MHC genes to limit antigen presentation is a key strategy for reducing host rejection of allogeneic CAR T cells. For example, current research demonstrated that B2M inactivation limits cell surface MHC class I (MHCI) expression by targeting β_2_-microglobulin (β_2_M) [68]. Finally, engineering allogeneic CAR cells with secondary ADRs may further mitigate graft rejection. For example, Mo et al. demonstrated that chimeric 4-1BBL-CD3ζ or OX40L-CD3ζ fusion constructs targeting 4-1BB and OX40 were undetectable on the surface of quiescent T cells and NK cells but were significantly upregulated following activation [69,70] (Figure 2G).

#### 3.1.7. Graft-Versus-Host Disease

Allogeneic CAR-T cell therapies carry risks of graft-versus-host disease (GVHD), where donor immune cells attack host tissues, which may become life-threatening in acute forms [71]. Strategies to mitigate GVHD include optimizing dosing regimens or selecting appropriate pre-infusion lymphodepleting agents to eliminate residual contaminating donor T cells from the infusion product. Second, CAR-T cell dose influences both efficacy and toxicity; dose adjustment can mitigate GVHD risk. For example, in paediatric allogeneic HSCT recipients, GVHD risk decreased following a dose of 5 × 10^4^ T cells/kg [72]. Furthermore, deletion of the αβTCR—a key mediator of alloreactivity—reduces rejection without compromising cytotoxicity [73,74] (Figure 2H).

#### 3.1.8. Systemic Toxicity

The robust efficacy of CAR-T cell therapy against drug-resistant and refractory diseases is offset by considerable toxicities [75]. Upon activation, CAR T-cells release abundant cytokines, leading to systemic cytokine toxicity. Consequently, all current CAR T-cell therapies incur significant toxicities, commonly manifested as fever, hypotension, and organ failure—specifically due to elevated inflammatory cytokine levels [76,77]. Some patients develop immune effector cell-associated neurotoxicity syndrome (ICANS), characterized by significant neurological symptoms such as aphasia, tremor, seizures, and headaches [78] (Figure 2I).

### 3.2. Advantages of CAR Cell-Derived Exosomes

#### 3.2.1. Higher Security

CAR-exosome therapy demonstrates improved safety compared with CAR-T cell therapy. First, CAR cell-derived exosomes exhibit low immunogenicity and avoid rejection [79]. Furthermore, CAR cell-derived exosomes are nanovesicles devoid of intact nuclear material, mitigating risks of insertional mutagenesis and oncogenesis associated with parental CAR cell therapy. CAR cell-derived exosomes exhibit reduced toxicity risk and contain readily cleared metabolites, thereby lowering systemic toxicities like cytokine release syndrome (CRS). Moreover, their generation from healthy donors supports potential off-the-shelf therapeutic applications [80]. Current study demonstrated that CAR T cell-derived exosomes inhibit tumor growth. Unlike CAR-T cells, CAR exosomes lack PD-1 expression; consequently, recombinant PD-L1 treatment did not inhibit their antitumor effects [81]. In an in vivo cytokine release syndrome model, CAR T-cell-derived exosomes exhibited a safer administration profile than CAR T therapy. Both 5 × 10^4^ CAR-T cells and 10 μg of CAR exosomes induced approximately 20% killing of 5000 tumor cells. Given that CAR-T cells contain 0.25–0.69 ng CAR protein per μg, corresponding to ~10 ng CAR protein per 5 × 10^4^ CAR-T cells, CAR exosomes achieved comparable cytotoxicity at lower CAR protein doses, thus enabling reduced dosage for enhanced therapeutic efficacy [82] (Figure 3A).

#### 3.2.2. Ability to Break the Blood-Brain Barrier

The blood-brain barrier is a highly organized multicellular structure comprising capillaries formed by tight junctions between cerebral microvascular endothelial cells, pericytes surrounding these capillaries, and astrocytes [83]. Impaired drug delivery across the blood-brain barrier complicates brain tumor therapy, underscoring the need for innovative treatment strategies [84]. Impaired drug delivery across the blood-brain barrier complicates brain tumor therapy, underscoring the need for innovative treatment strategies. While research functionalized exosomes with c(RGDyK) peptides to target ischemic brain damage, these exosomes showed no efficacy in targeting metastatic tumor cells [85]. Building on CAR-T cell therapy experience, CAR-T cell-derived exosomes have emerged as a novel approach for treating various cancers, including solid tumors. For example, Chang et al. conjugated single-chain variable fragments (scFv) to CAR NK cell-derived exosomes, retaining target cell killing ability while enhancing blood-brain barrier targeting [86] (Figure 3B).

#### 3.2.3. Multifunctional Loads and Efficient Delivery

CAR cell-derived exosomes effectively deliver chemotherapeutic drugs, nucleic acids, and immunomodulatory factors to synergistically enhance antitumor effects. For example, metastatic breast cancer patients with brain metastasis are often HER2-positive [87,88]. The multifocal and aggressive nature of breast cancer brain metastases is reflected in the limitations of surgical treatment [89]. Therefore, treatment for patients with HER2-positive breast cancer brain metastases primarily involves radiotherapy and systemic therapies. The significant adverse effects and variable efficacy of radiotherapy underscore the importance of drug therapy for breast cancer brain metastases (BCBM) [90]. However, poor tumor cell targeting and insufficient drug delivery imposed by the blood-brain barrier (BBB) are major obstacles to effective treatment of HER2-positive breast cancer brain metastases (BCBM) [91,92]. To address these challenges, researchers loaded micellar drugs onto CAR NK cell-derived exosomes (Exo^CAR/T7@Micelle^). Exo^CAR/T7@Micelle^ crossed the blood-brain barrier, selectively targeted HER2⁺ breast cancer cells, and enhanced therapeutic efficacy by disrupting ferroptosis defense through drug delivery [93] (Figure 3C).

#### 3.2.4. Regulation of the Tumor Microenvironment

The tumor microenvironment (TME), the site of tumor cell growth, comprises immune cells, stromal cells, the vascular system, and desmoplastic components [94]. Hypoxia, a key feature of the tumor microenvironment, enriches regions with components like CAFs and MDSCs that suppress T cell growth, activation, and cytotoxicity, thereby impairing antitumor immunity [95]. Cancer-associated fibroblasts (CAFs), major stromal components of the tumor microenvironment (TME), mediate critical processes including tumor proliferation, invasion, angiogenesis, inflammation, immunosuppression, and extracellular matrix (ECM) remodelling [96]. CAR cell-derived exosomes transfer nucleic acids and proteins from immune cells to recipient cells, modifying target cell function, remodeling the tumor microenvironment, and enhancing therapeutic efficacy [97,98]. For example, macrophages are abundant in the tumor microenvironment and are categorized as M1 macrophages (inhibiting tumor growth) or M2 macrophages (promoting tumor growth) [99]. Rao et al. developed biomimetic magnetic nanoparticles for cancer therapy by encapsulating them within gene-edited macrophage cell membranes. These nanoparticles upregulated SIRPα variants, inhibiting the CD44/SIRPα axis to promote M1 macrophage polarization [100]. Exosomes from M1 macrophages contain abundant pro-inflammatory cytokines, including TNF-α, IL-1β, IL-6, IL-12, and IL-23, which suppress cytotoxic Th1-type responses [101]. Thus, CAR exosomes can modulate tumor progression by regulating the tumor microenvironment (Figure 3D).

#### 3.2.5. Participation in the Diagnosis and Treatment of Cancer

CAR cell-derived exosomes function in tumor diagnosis and therapy. Multiple immune effector cells, including T cells, γδT cells, NK cells, NKT cells, and macrophages, have been engineered to express CARs and demonstrate antitumor activity in preclinical and clinical studies [102,103,104]. Exosomes from HER2-CAR-T cells carry HER2 proteins or associated RNAs, enabling early breast cancer diagnosis and treatment response monitoring [105]. In pancreatic cancer, CAR-T cell-derived exosomes carrying KRAS mutation-associated RNA or proteins may enable early diagnosis and recurrence monitoring through liquid biopsy detection [106]. Detection of PD-L1 levels in CAR-T cell-derived exosomes from melanoma patients can assess tumor immune escape status and guide immunotherapy [107] (Figure 3E).

## 4. CAR Cell-Derived Exosomes Could Serve as a Drug and Delivery Platforms for Tumor Therapy

### 4.1. Evolution of the CAR Structure

The 1970s discovery that cytotoxic T cells kill various tumor cells established their potential for cancer therapy [108]. Researchers conceptualized using T cells expressing chimeric antigen receptors (CARs) for tumor therapy in 1989 [108]. Up to now, the CAR structure has been optimized several times. First-generation CARs consisted of an scFv linked to the CD3ζ or FcεRIγ transmembrane and cytoplasmic ITAM domains. Although they can specifically target and kill tumor cells in mouse models, their low off-target cytotoxicity and lack of long-term persistence in vivo hinder clinical translation [109]. Subsequent research revealed that this resulted from the absence of essential co-stimulatory receptor ligands [110,111]. Therefore, in second-generation CARs, signaling domains from co-stimulatory receptors like CD28 or 4-1BB were incorporated between the transmembrane domain and the ITAM domain of CD3ζ [112]. However, despite the durable efficacy of CAR therapies against hematological malignancies, significant challenges remain in treating solid tumors. Fourth-generation CARs incorporate additional functionality beyond T cell activation, enabling localized secretion of cytokines and antibodies to modulate the solid tumor microenvironment. This targeted approach enhances efficacy while mitigating systemic adverse events [113,114,115]. The fifth-generation CAR is a versatile construct amenable to cost-effective mass production. Clinical studies and FDA approval of CAR-based products, resulting from sustained design innovation, confirm effective tumor targeting and safety, significantly advancing their clinical translation (Table 2).

### 4.2. CAR Cell-Derived Exosomes Are Spatially Targeted to Tumor Cells

CAR cell therapy is a form of engineered cellular immunotherapy wherein donor immune cells are genetically modified to express chimeric antigen receptors (CARs). These receptors enable the cells to target tumor cells, overcome the local immunosuppressive tumor microenvironment, and break immune tolerance [116]. The CAR molecule comprises three components: the extracellular domain, transmembrane domain, and intracellular domain [117]. The extracellular domain typically comprises a monoclonal antibody single-chain variable fragment that binds target antigens and a hinge region that links the components. The transmembrane domain anchors the CAR receptor to the cell membrane, ensuring stable expression. The intracellular domain includes a co-stimulatory domain and a signal transduction domain, which jointly mediate cellular activation. scFv specificity and affinity for tumor antigens determine CAR-T cell therapy safety and efficacy [117].

CAR cell-derived exosomes retain CAR, enabling more precise tumor-targeted drug delivery. For example, researchers developed CAR-T cell-derived engineered exosomes containing paclitaxel (PTX) to target mesothelin for lung cancer treatment. Results demonstrated that PTX@CAR-Exos target tumor cells, enabling selective drug accumulation and enhanced therapeutic efficacy while reducing off-target toxicity [118]. In conclusion, by altering the CAR structure, CAR cell-derived exosomes can be engineered to target specific cell types, resulting in high accumulation within target cells and low concentrations in surrounding healthy tissues.

### 4.3. CAR Cell-Derived Exosomes Target Tumor Sites Temporally

While CAR cell-derived exosomes precisely target tumor cells, ensuring efficient drug release and maximizing therapeutic efficacy upon target cell engagement remains challenging. Furthermore, hybrid therapeutic nanovesicles (hGLV), formed by fusing with drug-loaded thermosensitive liposomes, enable controlled release of exosomal antitumor drugs. For example, laser-irradiated hGLVs exhibited higher drug release at 1 and 6 h than non-laser-irradiated exosomes [119]. In conclusion, CAR cell-derived exosomes demonstrate controlled, time-specific delivery of therapeutic molecules, significantly enhancing antitumor efficacy and offering potential for advancing modern drug delivery systems.
ijms-26-07890-t002_Table 2Table 2Summary of advancements in research on CAR cell-derived exosomes in cancer.ClassificationTargetSubtypesResearch ProgressRefs.CAR-T cell-derived exosomesEGFRTNBCCetuzumab transduction shows TGI dose-dependence in the MDAMB-231 mouse xenograft model.[82]MSLNTNBCExosomes produced by CAR-T cells transduced with trastuzumab showed significant antitumor effects on the treatment of MCF-7 HER2 cells and SK-BR-3 cells.[82]MSLNLung cancerDelivery of PTX to tumor cells by continuous targeted delivery enhances antitumor effects and prolongs survival time of hormonal mice in CT-26 metastatic lung cancer model.[120]HER2HER-2-positiveExosomes from CAR-T cells targeting MSLN showed strong antitumor effects on MSLN-positive TNBC.[121]CAR NK cell-derived exosomesHER2HER-2-positiveAble to penetrate the blood-brain barrier and selectively exert antitumor effects on HER2-positive breast cancer cells in the brain.[21]CAR M cell-derived exosomesCXCL10lymphomaSignificantly enhances the immune activation and migration of T lymphocytes and promotes their differentiation into CD8+ T cells. Meanwhile, it increases the proportion of M1 macrophages, which exerts excellent antitumor activity in vivo.[122]

## 5. Combination Therapy Based on CAR Cell-Derived Exosomes

### 5.1. Chemotherapy

Chemotherapy, a common systemic treatment for tumors, often causes adverse events such as peripheral sensory neuropathy, cardiotoxicity, and myelosuppression due to its nonselective nature, compromising treatment efficacy [123,124,125,126]. CAR cell-derived exosomes enable precise tumor cell targeting; their combination with chemotherapy represents a promising therapeutic strategy. For example, researchers developed CAR-EDC, a novel drug coupler based on CAR-macrophage exosomes. CAR-M cell-derived exosomes enhance T lymphocyte activation, migration, and differentiation into CD8+ T cells [127]. Meanwhile, CAR exosomes from CAR-M cells contained elevated CXCL10. Furthermore, covalently conjugated SN38-loaded CAR exosomes underwent CAR-mediated endocytosis into Raji lymphoma cells, exerting significant antitumor effects attributable to both the chemotherapeutic activity of SN38 and the CXCL10-mediated antitumor immune response. In conclusion, this study demonstrates a novel strategy to enhance the antitumor effects of combined chemotherapy and CAR cell-derived exosomes [122] (Figure 4A).

### 5.2. Radiotherapy

The combination of radiation therapy and CAR cell-derived exosomes enhances target cell sensitivity to radiation. Radiation therapy is widely used and effective against most cancers [128]. However, inherent radiation resistance in colon and lung cancers remains a barrier to therapeutic efficacy, leading to poor outcomes [129,130]. The blood-brain barrier also prevents radiotherapy from exerting an anti-tumor effect in GBM [131]. Combining CAR-T cell-derived exosomes with radiotherapy mitigates radio-resistance, enhances blood-brain barrier (BBB) penetration, and increases drug accumulation at tumor sites. For example, inspired by CAR T cells, researchers engineered M1 macrophages with the exogenous fusion protein CAT-TMR-Anti-PD-L1-c-Myc (CAT-PD) and isolated their exosomes, termed DDRi@CAT-PD-M1Exos [132]. The engineered exosomes functioned as potent radiotherapy sensitizers by alleviating tumor hypoxia, inhibiting DNA damage repair, and remodeling the immunosuppressive microenvironment, thereby addressing three key limitations of radiotherapy [132]. Furthermore, M1 macrophage-derived exosomes reprogrammed M2 macrophages towards the M1 phenotype, enhanced their cytotoxicity against tumor cells in vitro, and promoted secretion of pro-inflammatory cytokines [133] (Figure 4B).

### 5.3. Immunotherapy

Immunotherapy utilizes active or passive mechanisms to induce an immune response that modulates tumor cell activity. Exosome properties originate from parental cells. Therefore, exosomes from CAR immune cells retain parent cell cytotoxicity while specifically targeting tumor cells. Monoclonal antibodies targeting specific molecules, which can be displayed on exosome surfaces, are powerful tools for activating antitumor immunity. PD-1 and PD-L1 are well-established immunosuppressive molecules; however, systemic administration of anti-PD-1 or anti-PD-L1 antibodies can trigger various immune-related adverse events (irAEs), including autoimmune disorders. Exosome-mediated delivery of PD1/PDL1 therapeutics enhances tumor suppression while reducing adverse effects [134]. Furthermore, CAR-T-derived exosomes demonstrate superior antitumor efficacy and reduced irAEs—particularly CAR-T-associated encephalopathy syndrome (CRES) and cytokine release syndrome (CRS)—compared to CAR-T cells, enhancing their clinical translational potential (Figure 4C).

### 5.4. Gene Therapy

Gene therapy introduces exogenous genes into target cells to correct, silence, or compensate for diseases caused by defective or aberrant genes [135]. However, limited control over the expression levels and localization of non-coding RNAs and CRISPR/Cas9 systems remains a major factor limiting the therapeutic efficacy of these gene therapy vectors for diverse diseases, including cancer. CAR cell-derived exosomes serve as carriers to deliver therapeutic molecules to the tumor site with precise spatiotemporal and dosage control [136,137]. For example, Wang et al. utilized CAR NK cell-derived exosomes to deliver miR-1249-3p, enabling precise SKOR1 targeting and regulating glucose homeostasis. This was achieved by modulating the SMAD6/MYD88/SMURF1 ternary complex and inhibiting the TLR4/NF-κB signalling pathway, thereby influencing disease progression [138]. researchers delivered miR-207 using NK cell-derived exosomes. By interacting with TLR4, miR-207 inhibited NF-κB signaling in astrocytes, thereby reducing pro-inflammatory cytokine release and alleviating depressive symptoms. Combining gene therapy with CAR-derived exosomes significantly enhances tumor therapy outcomes and facilitates clinical translation [139] (Figure 4D).

### 5.5. Photothermal Therapy

Photothermal therapy (PTT) employs heat generated by photothermal agents during NIR laser irradiation to ablate tumors and release drugs. However, photothermal therapy in the NIR-I window suffers from limited penetration depth and HSP-mediated thermotolerance [140,141,142]. The high temperatures required to effectively ablate tumor tissue and overcome HSP-induced thermal resistance inevitably damage adjacent normal organs [143]. Leveraging the targeting capability and ease of engineering of CAR T-cell-derived exosomes, researchers are developing novel combination therapies. For example, NK cells express Granzyme B, which induces apoptosis in target cells via perforin [144]. Researchers developed a therapeutic platform by fusing CAR-T cell-derived exosomes with photothermal-sensitive liposomes. Under laser irradiation, this platform demonstrated high efficacy with effective cascade tumor targeting and cytotoxicity, significantly enhancing anti-tumor efficiency in vivo [145]. In conclusion, combining CAR cell-derived exosomes with photothermal therapy (PTT) significantly enhances tumor penetration and reduces the required laser dose compared to traditional PTT, thereby improving therapeutic efficacy while minimizing adverse effects (Figure 4E).

### 5.6. Sonodynamic Therapy

Sonodynamic therapy (SDT) is a cancer treatment method using ultrasound and sonosensitizers [146]. Unlike PDT, SDT avoids limitations of penetration depth and collateral tissue damage, enabling treatment of deep-seated tumors using exosomes loaded with acoustic sensitizers [146]. However, acoustic kinetic therapy has limited efficacy against brain metastases due to the blood-brain barrier and tumor microenvironment [147,148]. CAR cell-derived exosomes efficiently penetrate the blood-brain barrier and precisely target cancer cells, addressing limitations of SDT monotherapy and demonstrating significant clinical translation potential. For example, Yuan et al. functionalized macrophage-derived exosomes with the AS1411 aptamer, catalase-loaded silica nanoparticles (CAT@SiO_2_), and the acoustic sensitiser ICG, demonstrating efficient blood-brain barrier penetration and tumor cell targeting. Furthermore, CAT-catalyzed oxygen generation from endogenous H_2_O_2_ alleviated tumor hypoxia, significantly enhancing SDT efficacy [149,150] (Figure 4F).

### 5.7. Photodynamic Therapy

Laser irradiation activates CAR cell-derived exosomes loaded with ferroptosis inducers and photosensitizers, achieving synergistic antitumor effects [151]. Photodynamic therapy (PDT) is a minimally invasive cancer treatment utilising reactive oxygen species (ROS) generated from photosensitisers [152]. Research demonstrates ferroptosis has significant therapeutic potential against many cancers, particularly apoptosis-resistant tumors. Ferroptosis is driven by the iron-dependent Fenton reaction and characterized by the accumulation of lipid peroxides [153]. For instance, using exosomes derived from CAR NK cells, researchers developed a novel biomimetic nanoplatform termed Exo^CAR/T7@Micelle^ [21]. This ROS-amplified micelle platform enables targeted therapy for HER2-positive breast cancer brain metastases (BCBM), disrupting the GPX4 ferroptosis defense via dual RSL3 and ROS action to significantly inhibit tumor growth. The nanobomb mPEG-TK-Ce6@RSL3, combined with photodynamic therapy, triggers a ROS amplification cascade and releases Ce6 upon micelle dissociation, achieving spatiotemporally controlled RSL3 release and reduced side effects [21] (Figure 4G).

### 5.8. Precision Therapy

Precision therapy involves designing drugs to target cancer-causing lesions at the cellular or molecular level. These drugs specifically enter target cells to perform functions such as gene silencing or compensation, without affecting other cellular phenotypes [154]. Significant advances have occurred in chimeric antigen receptor (CAR) T-cell therapy, with five products approved by the US Food and Drug Administration for treating hematological malignancies [82,118]. The CAR structure enables exosomes derived from CAR cells to target specific cells precisely and overcome certain limitations of CAR T-cell therapies. For example, researchers engineered NK cell-derived exosomes with a CAR structure. This overcame the blood-brain barrier, enabled spatiotemporally controlled release of therapeutic molecules, and enhanced anti-tumor efficacy by disrupting ferroptosis defence mechanisms [21].

## 6. Challenges and Outlook

Although CAR cell-derived exosomes hold considerable promise for clinical translation, several key challenges require resolution. First, production standardization remains a hurdle, as the yield and homogeneity of CAR cell-derived exosomes are influenced by cell sources, culture methods, and isolation techniques. Substantial batch-to-batch variability can compromise therapeutic efficacy. Thus, establishing reproducible cell cultivation protocols and standardized isolation workflows represents a critical unmet need. Additionally, scalable manufacturing of engineered CAR cell-derived exosomes faces significant limitations. Conventional laboratory-scale ultracentrifugation—when adapted for industrial production—incurs low recovery rates, high energy consumption, and vesicle membrane damage. Second, regarding safety, repeated administration poses a risk of immune responses despite the inherently low immunogenicity of exosomes. Furthermore, comprehensive clinical evidence regarding the impact of immunogenicity on therapeutic outcomes is still lacking. Third, the biodistribution pathways of exosomes in humans—including absorption, organ-specific accumulation, and excretion—are inadequately characterized. Longitudinal tracking is necessary to assess off-target toxicity, while the efficiency of targeted delivery to specific organs or cells requires further investigation.

In conclusion, while CAR cell-derived exosomes hold significant promise as cell-free therapeutic agents for drug delivery, regenerative medicine, and diagnostics, their clinical translation faces substantial challenges that must be overcome for broader implementation.

## 7. Conclusions

CAR-T cell therapy has revolutionized the treatment of malignant hematologic malignancies and accelerated the development of novel targeted cellular therapies. However, efficacy is limited by barriers, including the tumor microenvironment (TME) and host-mediated graft rejection. Given the properties of exosomes, CAR cell-derived exosomes represent a promising solution to these challenges. Research demonstrates that CAR cell-derived exosomes possess therapeutic potential and act as effective drug carriers. Combining them with other cancer treatments enhances efficacy while reducing side effects. Meanwhile, CAR cell-derived exosome therapy overcomes barriers such as the TME and blood-brain barrier that challenge CAR-T cell therapy, serving as an important complement to it. In conclusion, CAR cell-derived exosome therapy provides a foundation for combination tumor therapy.

## Figures and Tables

**Figure 1 ijms-26-07890-f001:**
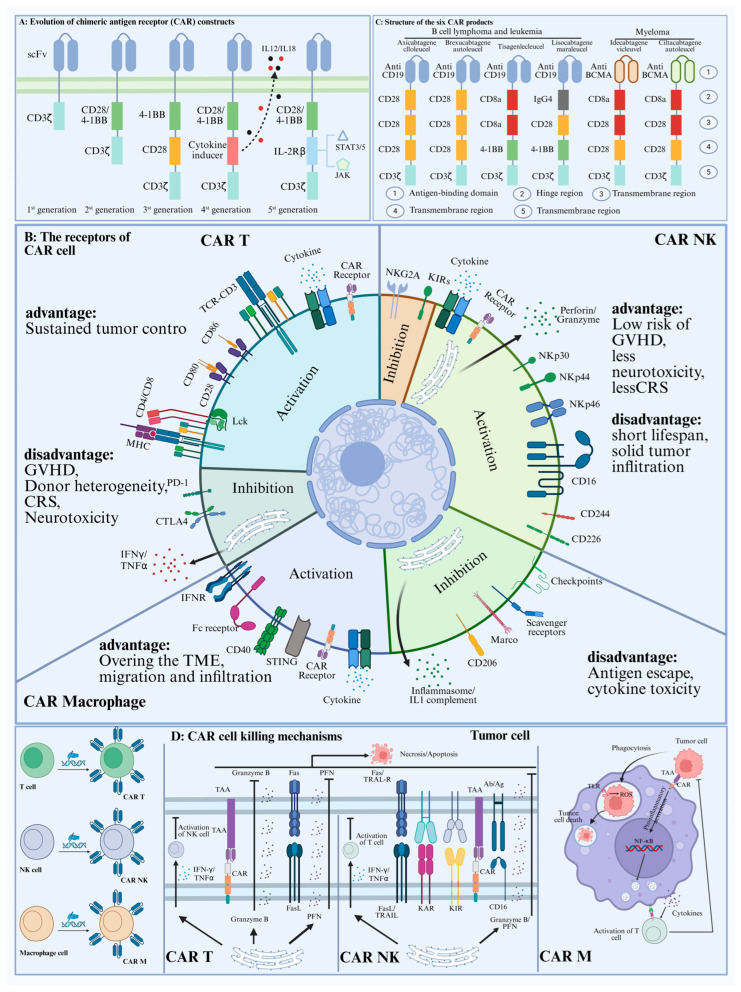
CAR cell-derived exosomes could serve as drug and delivery platforms for tumor therapy. (**A**) Evolution of the CAR structure. CAR engineering has evolved through five generations of structural optimization, each enhancing functional specificity, persistence, and safety. Initial designs (1st-gen) relied solely on CD3ζ for T-cell activation, exhibiting limited efficacy. Subsequent integration of co-stimulatory domains (e.g., CD28 or 4-1BB) in 2nd-gen CARs markedly enhanced T-cell persistence and tumor clearance. Third-generation constructs combined dual co-stimulatory motifs to amplify signaling potency. Fourth generation includes logic-gated systems for precision targeting, armored CARs modifying the tumor microenvironment, and suicide switches augmenting safety. This structural evolution continues to address challenges in specificity, durability, and toxicity management. Building upon fourth-generation CAR designs, fifth-generation CARs incorporate additional signaling activation and microenvironment regulatory modules to address core solid tumor challenges, including the immunosuppressive microenvironment, T-cell exhaustion, and on-target/off-tumor toxicity. (**B**) The active and inhibitory receptors of CAR cells. CAR-T, CAR-NK, and CAR-M cells exhibit distinct advantages and limitations. Advantages of CAR-Based Therapies in Cancer Treatment (advantages): CAR-T cells can achieve long-term tumor control; CAR-NK cells exhibit a lower risk of GVHD, CRS and neurotoxicity compared to CAR-T cells; CAR-M cells demonstrate the ability to overcome limitations imposed by the tumor microenvironment. Disadvantages of CAR-Based Therapies in Cancer Treatment (disadvantages): CAR-T cell therapy is associated with significant risks, including CRS, neurotoxicity, GVHD and challenges related to donor heterogeneity; CAR-NK cell therapy faces limitations such as shorter persistence in vivo and restricted tumor infiltration; CAR-M cell therapy is susceptible to tumor immune escape mechanisms, potentially compromising treatment efficacy. (**C**) Structure of six FDA-approved CAR cell drugs. Currently, six CAR cell products are approved by the US FDA for treating B-cell lymphoma, B-cell leukemia, and multiple myeloma. These therapies differ structurally, particularly in their antigen-binding domains, hinge regions, transmembrane regions, co-stimulatory domains, and T-cell activation domains. (**D**) Tumor-killing mechanism of CAR cells. CAR cells kill tumor cells by inducing apoptosis upon recognizing and binding target antigens.

**Figure 2 ijms-26-07890-f002:**
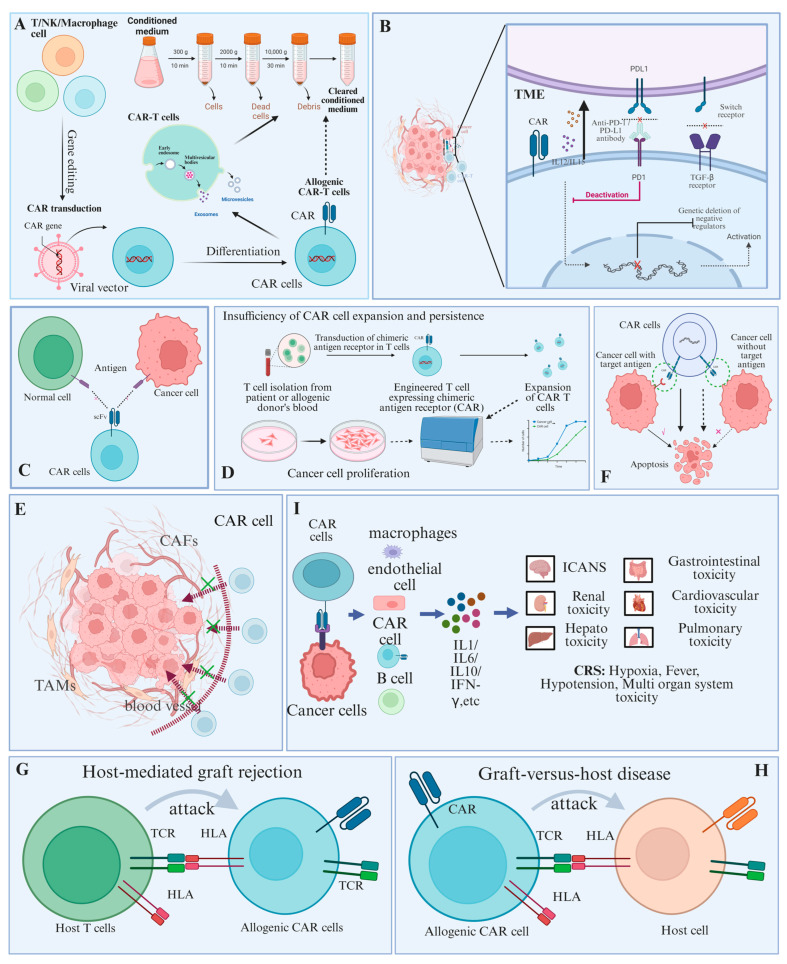
Limitations of CAR cell therapy. (**A**) Construction of CAR cells and extraction of exosomes. (**B**) The malignant tumor microenvironment hinders the contact of CAR cells with tumor cells. (**C**) CAR cells have fewer selectable targets. (**D**) Insufficiency of CAR T cell expansion and persistence. (**E**) Inefficiency of CAR T cell trafficking and infiltration into tumors. (**F**) CAR cells are susceptible to antigen escape. (**G**) CAR cells induce host-mediated immune rejection. (**H**) CAR cells can cause GVHD. (**I**) CAR cells carry the risk of causing a cytokine storm.

**Figure 3 ijms-26-07890-f003:**
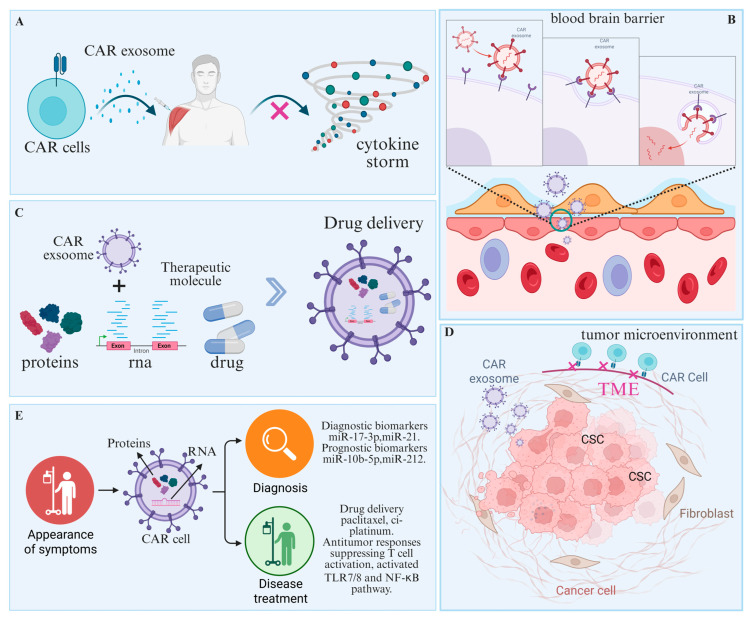
Advantages of CAR cell-derived exosomes. (**A**) CAR cell-derived exosomes have a higher safety profile. (**B**) CAR cell-derived exosomes were capable of breaching the blood-brain barrier. (**C**) CAR cell-derived exosomes can be used to load and deliver therapeutic molecules. (**D**) CAR cell-derived exosomes are capable of infiltrating the tumor microenvironment. (**E**) CAR cell-derived exosomes can serve as diagnostic and therapeutic molecules for tumor therapy.

**Figure 4 ijms-26-07890-f004:**
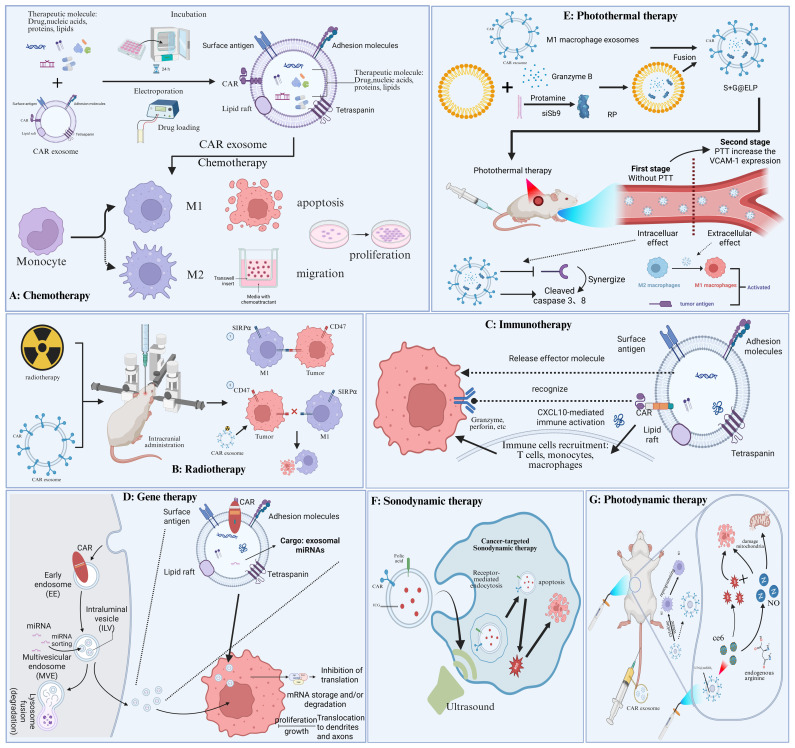
Combination therapy based on CAR cell-derived exosomes. (**A**) The combination of CAR cell-derived exosomes with chemotherapy enhances the sensitivity of tumor cells to the drug. (**B**) The combination of CAR cell-derived exosomes and radiotherapy enhances tumor recognition and antitumor effects. (**C**) The combination of CAR cell-derived exosomes and immunotherapy can better mobilize the body to recognize and fight tumor cells. (**D**) CAR cell-derived exosomes can carry therapeutic miRNAs into recipient cells and perform gene silencing functions. (**E**) CAR cell-derived exosomes make photothermal therapy more targeted with fewer side effects. (**F**) CAR cell-derived exosomes enhance the targeting of sonodynamic therapy. (**G**) The combination of CAR cell-derived exosomes and photodynamic therapy promotes the recognition and killing of tumor cells.

## Data Availability

Not applicable.

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
