# Peer review of "CAR Cell-Derived Exosomes in Cancer Therapy: Biogenesis, Engineering Strategies and Antitumor Mechanisms"

_ijms, 2025, doi:10.3390/ijms26167890_

Round 1
Reviewer 1 Report
Comments and Suggestions for Authors
Authors have made great efforts in making a compelling case in favor of the CAR exosomes. While the review is well written and there are several artworks, the scaling of those poster size artwork in half a page is not fair to the artwork and also not very legible. Authors can either decide to break the artworks in more figures or remove a few sections from each figure to focus on the most important ones so that they can be read easily with clarity.
Reviewer 2 Report
Comments and Suggestions for Authors
This manuscript, titled “CAR cell-derived exosomes in cancer therapy: current applications, therapeutic advances, and potential” by Chaohua Si et al., provides a comprehensive review of the emerging use of chimeric antigen receptor (CAR) cell-derived exosomes in cancer treatment. It highlights the limitations of traditional CAR therapies (CAR-T, CAR-NK, and CAR-M) and explores how exosomes could address these issues by improving safety, delivery, and interaction with the tumor microenvironment. The review includes an extensive literature review, offering valuable mechanistic insights and preclinical evidence that support the potential clinical use of CAR-derived exosomes. Although the topic is highly relevant, the current manuscript requires significant revision to increase its chances of publication in IJMS. It has notable issues with language clarity, organization, and formatting. Addressing these areas will greatly enhance the quality and impact of the review. Below are my specific comments that might help the authors to improve their review here:
- The manuscript is dense and could benefit from clearer logical flow. Reorganizing these sections may improve clarity for readers interested in the topic. I suggest adding subheadings and splitting the content into shorter sections, rather than keeping it in its current long format.
- The data in the key tables and figures is not very clear. All three figures are too crowded, and splitting them into sections could improve clarity. The authors mention clinical trials and studies; however, they do not include relevant clinical trial IDs in the table. Please consider adding that essential information in Table 1.
- There are many repetitive sections in this manuscript that unnecessarily lengthen it. For example, limitations of CAR-T (such as TME, antigen escape, and trafficking issues) are repeated in different sections. The authors should think about summarizing this in a single dedicated section instead of repeating it multiple times.
- The discussion section lacks appropriate future perspectives and improvements that will occur in the CAR exosomes field. I encourage the authors to consider this and improve the discussion section accordingly.
Overall, the manuscript is well-crafted and backed by a broad selection of up-to-date and pertinent references, boosting its depth and value. The explanation of how CAR-exosome technology integrates with different treatment methods (such as chemotherapy, radiotherapy, immunotherapy, gene therapy, etc.) is clear and innovative. I suggest that the authors correct grammatical mistakes and condense overly long sections to enhance the overall effectiveness.
Comments on the Quality of English LanguageThe English language is fine, but there are issues with lengthy sentences, repetitiveness, frequent grammatical errors, and inconsistent terminology. It needs improvement before publication.
Reviewer 3 Report
Comments and Suggestions for Authors
The review in the title and abstract aims to „review the roles, mechanisms and current research status of CAR cells and their exosomes in tumor therapy “. While this is an interesting topic that deserves to be reviewed. However the manuscript itself is lacking focus.
The review starts with introduction about combination of CAR expressing cells with exosomes. However subsequent paragraphs are description of chosen aspects of current CAR T, NK and macrophages therapy. This section ends with figure.1 that is not clearly described and is confusing, especially part B that contains schematics of different proteins from immune cells, with “advantages and disadvantages” of those without any clear comment. Moreover, the quality of attached figures makes it somehow unreadable.
Subsequent Table 1: Part of published clinical trials of CAR cell therapies, contains information about chosen clinical trials with no clear key (in line 97 the authors mentioned that there are more than 1000 clinical trials). Moreover, in the column disrupted genes, some have CD52 and TRAC, and some has TALEN which is the name of nuclease, not human gene.
More misleading fragments occurred, such as in line 325 “Most of the current CAR cell products are allogeneic products “, which is simply not true.
Therefore, I would strongly suggest for authors to perform a major revision of the manuscript and focus only on one aspect, in.ex. “CAR cell-derived exosomes” from the title. SSuch specific topic - (CAR cells exosomes) should be consistently followed throughout the article.
Comments on the Quality of English Language
I would strongly suggest submitting the manuscript to professional language edition before re-submission.
Round 2
Reviewer 2 Report
Comments and Suggestions for Authors
The authors have thoroughly addressed all my previous concerns through their revisions, and the revised manuscript is very well done.
Reviewer 3 Report
Comments and Suggestions for Authors The authors added the description of the figures; however, I am still not sure how they are connected with exosomes that they promised to write about in the title and abstract. The table 1 with selectively chosen clinical trials also seems not to be connected to the topic at all. It contains trials for allogeneic products with some mistakes and unclear information such as “site” column or the TALEN, between disrupted genes. On top of that we have a long section 3.1 with general information about CAR T-cell therapy limitations that is really general and not connected to exosomes. Moreover, in the section about exosomes itself, many references are not adequate. In.ex. Line 293 “Moreover, their generation from healthy donors supports potential off-the-shelf therapeutic applications [102].” -reference 102 (Teachey et al 2016) is about predicting CRS in CD19 CAR treated patients - not about generation of exosomes/CAR T-cells from healthy patients. Line 358 “Exosomes from HER2-CAR-T cells carry HER2 proteins or associated RNAs, enabling early breast cancer diagnosis and treatment response monitoring [128].” The reference 128 (Guo et al 2024) the authors did their studies on HER2-CAR T-cells, not exosomes derived from it. Also the authors suggested that newly identified genes might be a theoretical basis for improving treatment, not the diagnosis. Line 360-362 “In pancreatic cancer, CAR-T cell-derived exosomes carrying KRAS mutation associated RNA or proteins may enable early diagnosis and recurrence monitoring through liquid biopsy detection [129].” The reference 129 (Ramalingam et al 2024) is about developing a dual targeting (MSLN & CEA) CAR protein towards KRAS-mutated PDAC. It does not mention exosomes or diagnosis.Author Response
Please see the attachment.
